# RepMedSAM: Segment Anything in Medical Images with Lightweight CNN

Zehan Zhang[1][0000−0002−0451−985X], Rui Huang[1], and Ning Huang[1]

Hangzhou Genlight Medtech Co. Ltd., Zhejiang, China
`zzehan0123@gmail.com, huangn@gmail.com`

**Abstract.** Traditional deep learning segmentation models require designing network structures and loss functions specific to different tasks, followed by training dedicated models, which leads to a significant amount of repetitive work. The Segment Anything Model (SAM) provides a unified framework for handling segmentation tasks. However, the current SAM model is mainly applicable to natural images and may require substantial computational resources during inference, posing challenges for widespread clinical implementation. In this work, we utilize RepViT as the Image Encoder to develop a lightweight SAM structure. The training phase consists of two main parts: knowledge distillation and fine-tuning. During the inference phase, reparameterization is employed to optimize inference speed. The proposed method achieves an average DSC of 0.8688 and an average NSD of 0.8746 on the validation set, and it improves inference speed while increasing the number of parameters compared to the baseline.

**Keywords:** Segment Anything Model · Medical image · Lightweight · Knowledge distillation.

## 1 Introduction

Medical image segmentation is a crucial component in clinical practice, where the accuracy of segmentation results is essential for ensuring the safety and efficacy of medical diagnosis and treatment. Existing methods typically design network structures and loss functions tailored to specific tasks and train dedicated models, resulting in poor generalization performance across different segmentation tasks and modalities.

Recently, various foundational models have garnered significant attention from researchers in the field of computer science due to their outstanding performance. In natural images, the Segment Anything Model (SAM) [1] performs interactive segmentation using prompts such as points, boxes, masks, and text. The introduction of such prompt engineering enables SAM to adapt to nearly all downstream segmentation tasks, achieving impressive results comparable to models specifically trained for particular tasks. However, unlike natural images, medical images comprise multiple modalities with significant differences between

them. Therefore, the performance of the foundational SAM model in medical image segmentation is quite limited. Works like MedSAM [3] have successfully transferred SAM to the medical image segmentation domain, achieving excellent performance. Nonetheless, these models generally require substantial computational resources during inference, and their inference speed does not meet the real-time requirements of clinical applications. Hence, developing a lightweight SAM for efficient deployment in medical image segmentation is a meaningful and promising research area.

Currently, many lightweight SAM models have emerged in natural images to enable these applications to run on resource-constrained terminal devices. Mobile-SAM [6] replaces SAM's image encoder with a lighter structure and completes training within a day using a decoupled distillation approach. Compared to SAM, MobileSAM achieves comparable performance with 60 times fewer parameters and can run stably on a CPU. EfficientViT-SAM [6] proposes a pre-training framework called SAMI with masked images, which significantly enhances the performance of image mask pre-training methods and extends well to various tasks such as image classification, object detection, and semantic segmentation. Compared to the original SAM, EfficientSAM reduces the number of parameters by 20 times, accelerates the running speed by 20 times, and surpasses models like MobileSAM and FastSAM [7] in performance. RepViT [4], which incorporates re-parameterized convolutions into the MobileNetV3 [2] architecture, forms a lightweight CNN resembling a ViT structure. When used as the image encoder and combined with SAM, it achieves faster and better results than MobileSAM.

In this paper, we build upon RepViT and SAM, implementing several minor architectural and preprocessing adjustments to significantly reduce the overall model parameters and computational load. The model undergoes training in two stages: distillation of the image encoder and overall fine-tuning. During the inference phase, we leverage structural re-parameterization to further reduce the number of parameters and enhance speed, all while maintaining accuracy.

## 2   Method

We propose a lightweight foundational model for medical image segmentation based on SAM and RepViT. The details of the proposed method are described as follows.

### 2.1   Preprocessing

We primarily follow the baseline data processing approach for three-channel 2D data. The inference phase consists of the following steps:

- Resize the longest edge to 256 while maintaining the aspect ratio of the image.
- Normalize the image pixel value to the [0, 1] range using Max-Min Normalization.

– Pad the image to [256, 256].
– Align the coordinates of the box with the resized image.

For 3D data preprocessing during the inference phase, we perform slice-by-slice operations for each box. Each slice is expanded to three channels, and the data preprocessing method then follows the same procedure as described above for 2D data.

During the training phase, to avoid excessive disk usage, we directly read npz files for training. For 3D data, one random slice in the stack is read per iteration. To utilize more slices in the same stack, one 3D data is read multiple times during each training epoch.

### 2.2   Proposed Method

The overall architecture of the proposed method is identical to MedSAM, as illustrated in Figure 1. It mainly consists of three parts: the Image Encoder, the Mask Decoder, and the Prompt Encoder. For each box, the method predicts the corresponding mask individually.

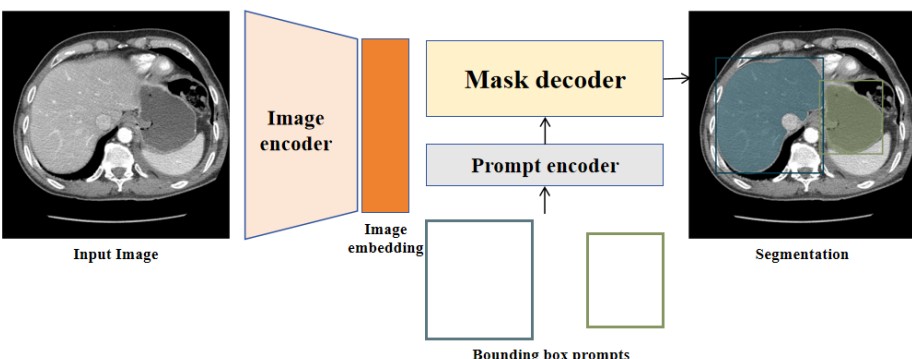

**Fig. 1.** The network architecture of MedSAM.

We replace the Image Encoder with the lightweight RepViT model, the overall structure of which is shown in Figure 2. This structure comprises multiple stacked RepViTBlocks.

The structure of a single RepViTBlock is shown in Figure 3. Figure 3(a) depicts the structure of the MobileNet Block. The RepViT Block is an improved version based on this structure. Its architecture during training and inference is illustrated in Figure 3(b).

Additionally, the training of RepSAM is divided into two stages. The first stage involves knowledge distillation for the Image Encoder. In this stage, instead of using a larger parameter Image Encoder as the teacher model, we directly use the pre-trained TinyViT. Despite the smaller parameter count of the teacher

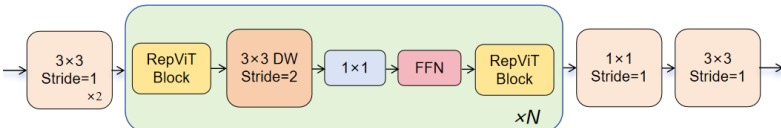

**Fig. 2.** The network architecture of RepViT.

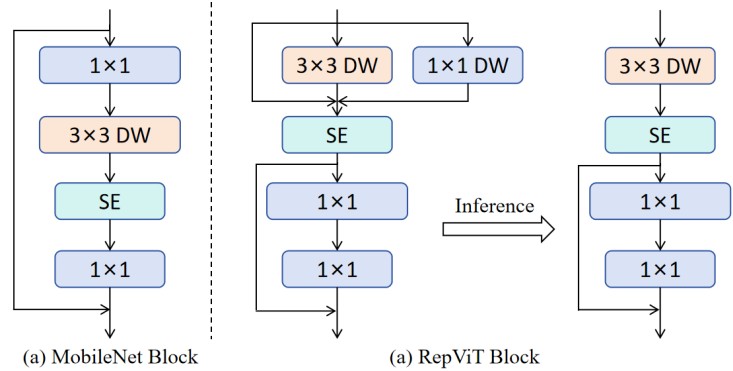

(a) MobileNet Block          (a) RepViT Block

**Fig. 3.** The structure of a single RepViT Block

model, the RepViT obtained by distillation learning achieves better results compared to training from scratch. The second stage involves fine-tuning the overall structure of RepSAM.

Loss function: We employ KLDivLoss as the loss function for the first stage of knowledge distillation. During the fine-tuning stage, the loss function is a combination of DiceLoss, BCELoss, and MSELoss.

### 2.3   Post-processing

We maintain the same approach as the baseline by post-processing the predicted masks through cropping and resizing to align the results with the input images.

## 3   Experiments

### 3.1   Dataset and evaluation measures

We use the dataset provided by the challenge, along with an additional public dataset, ToothSeg, for model training. Furthermore, during the distillation stage of the Image Encoder, we also include the validation set provided by the challenge in the training process. Throughout training, we maintain the data format as .npz. For 3D data, a random slice is read in each iteration, and during each epoch, each 3D dataset is traversed multiple times to read and train on multiple slices from the same data.

The evaluation metrics include two accuracy measures—Dice Similarity Coefficient (DSC) and Normalized Surface Dice (NSD)—alongside one efficiency measure—running time. These metrics collectively contribute to the ranking computation.

## 3.2   Implementation details

**Environment settings** The development environments and requirements are presented in Table 1.

**Table 1.** Development environments and requirements. (mandatory table)

| | |
|---|---|
| System | Ubuntu 18.04.6 LTS |
| CPU | Intel(R) Core(TM) i9-10900K CPU @ 3.70GHz |
| RAM | 64GB |
| GPU (number and type) | One NVIDIA GeForce RTX 3090 24G |
| CUDA version | 11.8 |
| Programming language | Python 3.9 |
| Deep learning framework | PyTorch(torch 2.1.2, torchvision 0.16.2) |
| Code | |

**Training protocols** The training process is divided into two stages. The first stage involves the distillation learning of the Image Encoder. The training protocol for this stage is shown in Table 2. In this stage, RepViT learns relevant knowledge from the pre-trained TinyViT and encodes the images. We use KLD-Loss as the loss function. Through knowledge distillation, RepViT can quickly converge within 10 epochs, and we select the best distillation model based on the loss value.

**Table 2.** Training protocols. (mandatory table)

| | |
|---|---|
| Pre-trained Model | MedSAM [3] |
| Batch size | 4 |
| Patch size | 256×256×3 |
| Total epochs | 10 |
| Optimizer | AdamW |
| Initial learning rate (lr) | 1e-4 |
| Lr decay schedule | CosineAnnealingLR |
| Loss function | KLDivLoss |

The second stage involves fine-tuning the entire RepSAM. First, we load the pre-trained weights for the three components: the Image Encoder loads the

weights from the knowledge-distilled RepViT, and the Prompt Encoder and Mask Decoder directly load the provided baseline weights. The remaining training protocol details are shown in Table 3, with model parameters and computational load only calculated for the Image Encoder. During the training phase, we apply a random jitter of 5 pixels to the input boxes and randomly flip the images for augmentation. Additionally, we employ the EMA (Exponential Moving Average) strategy to enhance the model's robustness. The data is split into training and validation sets in a 4:1 ratio. The optimal model is selected based on the dice metric evaluated on the validation set.

**Table 3.** Training protocols. (mandatory table)

| | |
|---|---|
| Pre-trained Model | MedSAM [3] RepViT [4] |
| Batch size | 4 |
| Patch size | $256\times256\times3$ |
| Total epochs | 30 |
| Optimizer | AdamW |
| Initial learning rate (lr) | 2e-5 |
| Lr decay schedule | CosineAnnealingLR |
| Training time | 61 hours |
| Loss function | DiceLoss + BCELoss + MSELoss |
| Number of model parameters | 23.16M[1] 23.04M(inference) |
| Number of flops | 7.23G[2] 7.08G(inference) |
| $CO_2$eq | 16.8Kg[3] |

## 4    Results and discussion

### 4.1    Quantitative results on validation set

We compare our method with the baseline, MedSAM, and other models. The quantitative evaluation results are shown in Table 4.1, listing the DSC and NSD for nine modalities, along with the average DSC and NSD across all modalities. Our proposed method achieves an average Dice of 0.8688 and an average NSD of 0.8746 on the validation set. Additionally, we compare our method with RepMedSAM trained from scratch. The table shows that RepMedSAM trained from scratch already achieves results surpassing both the baseline and the larger MedSAM. Moreover, incorporating knowledge distillation in the initial phase significantly enhances the final segmentation results.

The proposed method achieves good segmentation results across various modalities, including CT, MR, US, X-Ray, Dermatology, Endoscopy, and Fundus. However, its performance is relatively poorer on PET and Microscopy, possibly due to the data quality and the inherent difficulty of these segmentation

tasks. Additionally, there is nearly a 10% difference in the DSC metric for US segmentation between RepMedSAM and the baseline, which we suspect is due to differences in the training data. Furthermore, incorporating the ToothSeg dataset into the training process significantly improves the segmentation results for X-Ray in the validation set.

**Table 4.** Quantitative evaluation results.

| Target | Baseline | | MedSAM | | RepMedSAM | | RepMedSAM(distill) | |
|---|---|---|---|---|---|---|---|---|
| | DSC(%) | NSD(%) | DSC(%) | NSD(%) | DSC(%) | NSD(%) | DSC(%) | NSD(%) |
| CT | 89.10 | 91.03 | 90.40 | 92.09 | 86.30 | 89.19 | 93.88 | 96.36 |
| MR | 83.28 | 86.09 | 84.51 | 86.68 | 80.38 | 83.41 | 87.86 | 91.73 |
| PET | 55.10 | 29.12 | 62.42 | 47.64 | 67.06 | 50.25 | 64.41 | 41.30 |
| US | 94.77 | 96.81 | 91.92 | 95.55 | 84.39 | 89.36 | 84.46 | 89.46 |
| X-Ray | 75.82 | 80.38 | 78.28 | 84.01 | 85.06 | 90.37 | 85.92 | 91.19 |
| Dermotology | 92.47 | 93.86 | 91.37 | 92.81 | 94.55 | 96.02 | 94.70 | 96.15 |
| Endoscopy | 96.04 | 98.11 | 96.83 | 98.86 | 94.80 | 97.35 | 96.50 | 98.66 |
| Fundus | 94.81 | 96.42 | 95.01 | 96.64 | 95.38 | 96.96 | 94.54 | 96.22 |
| Microscopy | 61.63 | 65.39 | 67.90 | 74.65 | 75.52 | 82.29 | 79.65 | 86.08 |
| Average | 82.56 | 81.91 | 84.30 | 85.44 | 84.83 | 86.13 | 86.88 | 87.46 |

### 4.2    Qualitative results on validation set

Figure 4 shows the visual comparison of the proposed method and the baseline on selected data. It can be observed that RepMedSAM, trained in two stages of knowledge distillation and fine-tuning, achieves the best segmentation results on these datasets.

Figure 5 illustrates the segmentation results of the proposed method across various modalities. It is evident from the figure that the segmentation results in the left four columns are outstanding, while those in the right two columns are relatively mediocre.

### 4.3    Segmentation efficiency results on validation set

Table 5 presents the runtime comparison between the proposed method and the baseline on selected validation sets. All times are measured from tests conducted on a local machine CPU. Additionally, we compare the runtime before and after re-parameterization.

### 4.4    Results on final testing set

### 4.5    Limitation and future work

During this training process, to reduce the training time, we only included an additional X-Ray dataset for training to improve the segmentation accuracy

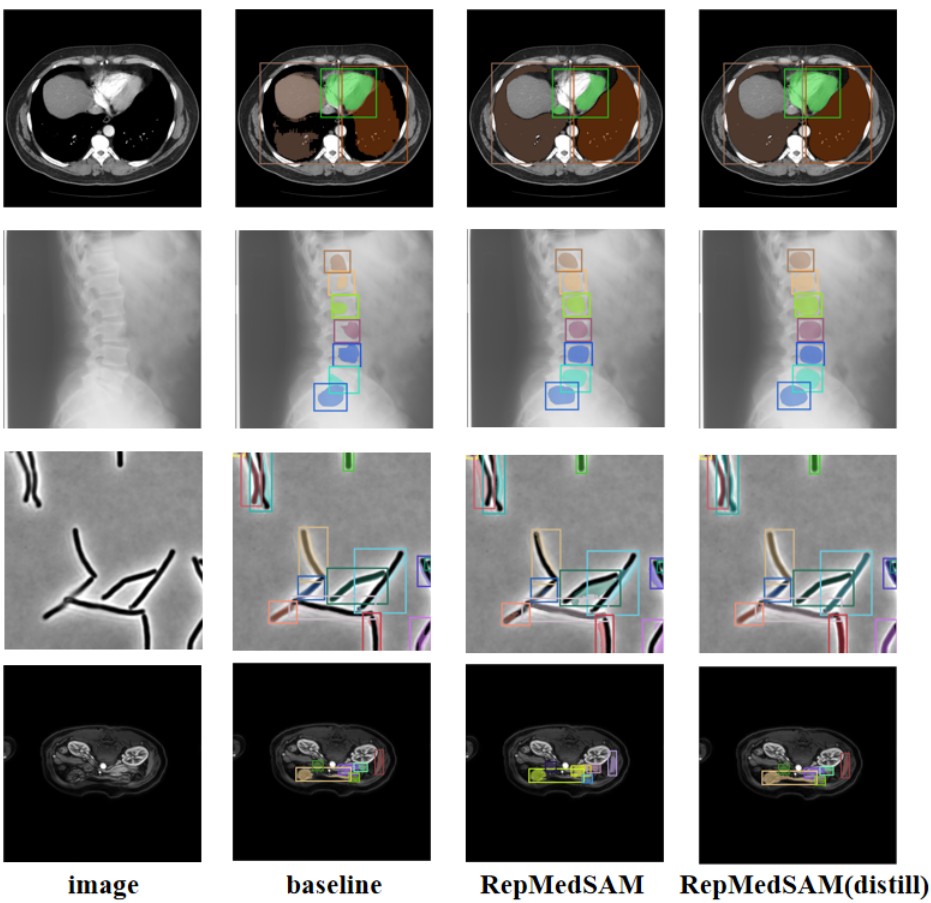

**Fig. 4.** Comparison between segmentation results of different methods.

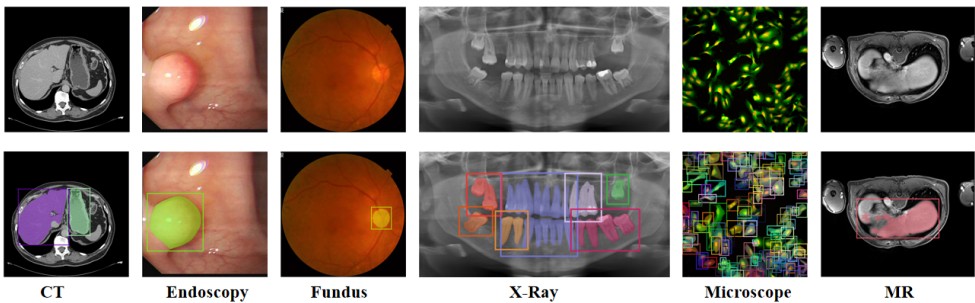

**Fig. 5.** Segmentation Results for Specific Modalities.

**Table 5.** Quantitative evaluation of segmentation efficiency in terms of running time (s).

| Case ID | Size | Num. Objects | Baseline | w/o rep | RepMedSAM |
|---|---|---|---|---|---|
| 3DBox_CT_0566 | (287, 512, 512) | 6 | 296.77 | 241.14 | 170.87 |
| 3DBox_CT_0888 | (237, 512, 512) | 6 | 79.30 | 64.61 | 45.88 |
| 3DBox_CT_0860 | (246, 512, 512) | 1 | 10.95 | 8.98 | 6.48 |
| 3DBox_MR_0621 | (115, 400, 400) | 6 | 124.53 | 101.81 | 72.35 |
| 3DBox_MR_0121 | (64, 290, 320) | 6 | 80.05 | 65.18 | 45.53 |
| 3DBox_MR_0179 | (84, 512, 512) | 1 | 10.62 | 8.80 | 6.28 |
| 3DBox_PET_0001 | (264, 200, 200) | 1 | 6.69 | 5.48 | 3.98 |
| 2DBox_US_0525 | (256, 256, 3) | 1 | 0.54 | 0.44 | 0.32 |
| 2DBox_X-Ray_0053 | (320, 640, 3) | 34 | 1.47 | 1.32 | 1.17 |
| 2DBox_Dermoscopy_0003 | (3024, 4032, 3) | 1 | 0.80 | 0.70 | 0.52 |
| 2DBox_Endoscopy_0086 | (480, 560, 3) | 1 | 0.55 | 0.44 | 0.32 |
| 2DBox_Fundus_0003 | (2048, 2048, 3) | 1 | 0.60 | 0.48 | 0.36 |
| 2DBox_Microscope_0008 | (1536, 2040, 3) | 19 | 1.15 | 1.04 | 0.90 |
| 2DBox_Microscope_0016 | (1920, 2560, 3) | 241 | 8.55 | 8.44 | 8.39 |

on the validation set. Including more datasets for training could enhance the model's generalization ability further. Additionally, we attempted to use the ONNX engine for inference, but the runtime did not show significant reduction. Further work on deployment could focus on reducing the required inference time.

## 5    Conclusion

We combine the lightweight CNN structure, RepViT, with SAM and apply it to medical image segmentation. Through training in two stages, knowledge distillation and fine-tuning, RepMedSAM achieves higher segmentation accuracy and faster segmentation speed compared to the baseline.

**Acknowledgements**  We thank all the data owners for making the medical images publicly available and CodaLab [5] for hosting the challenge platform.

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

**Table 6.** Checklist Table. Please fill out this checklist table in the answer column.

| Requirements | Answer |
| --- | --- |
| A meaningful title | Yes |
| The number of authors ($\leq 6$) | 3 |
| Author affiliations and ORCID | Yes |
| Corresponding author email is presented | Yes |
| Validation scores are presented in the abstract | Yes |
| Introduction includes at least three parts: background, related work, and motivation | Yes |
| A pipeline/network figure is provided | Figure 2 |
| Pre-processing | Page 2 |
| Strategies to data augmentation | Page 3 |
| Strategies to improve model inference | Page 3 |
| Post-processing | Page 4 |
| Environment setting table is provided | Table 1 |
| Training protocol table is provided | Table 2 3 |
| Ablation study | Page 7 |
| Efficiency evaluation results are provided | Table 5 |
| Visualized segmentation example is provided | Figure 4 5 |
| Limitation and future work are presented | Yes |
| Reference format is consistent. | Yes |
| Main text $>= 8$ pages (not include references and appendix) | Yes |