# OpenReview forum: "RepMedSAM: Segment Anything in Medical Images with Lightweight CNN"
_thecvf.com/CVPR/2024/Workshop/MedSAMonLaptop — CVPR24 MedSAMonLaptop_

### Official Review · Reviewer_rJs7 · 2024-06-10
**It's good, but there are a few things missing**

**Rating:** 6
**Confidence:** 5

**Review:**

Need to explain in details why with distillation, the results improved with that large margin, which much higher than the baseline.

---

### Official Review · Reviewer_XXEb · 2024-06-11
**This paper presents an approach to medical image segmentation using a lightweight convolutional neural network (CNN) architecture. The authors propose combining the Segment Anything Model (SAM) with RepViT, a vision transformer-based model, to create a more efficient and effective segmentation method. The paper demonstrates significant improvements in segmentation accuracy and speed over existing models.**

**Rating:** 7
**Confidence:** 3

**Review:**

## Pros:
1. This paper combined MedSAM and CNN-based transformer RepViT to enhance the performance and efficiency. It first distilled a RepViT
image encoder from pretrained TinyViT image encoder, and then it fintuned the entire pipeline. It improves the efficiency from about 300 seconds to 170 seconds on the most complex 3D case (CT_0566). For the performance, it achieved the DSC of 0.8688 and NSD of 0.8746, outperforming the baseline and other models.

## Cons:
1. This paper doesn't apply any other tools to improve the performance of deployments (e.g. torch script, torch.jit, onnx, etc), which may further improve the efficiency.
2. Some minor problems in the paper writing (e.g. the phrases "mandatory table"  in Table 1 should be deleted).

---

### Official Review · Reviewer_1FfH · 2024-06-12
**Well-written paper with good results but no code**

**Rating:** 9
**Confidence:** 3

**Review:**

Pros:
- The results show that utilizing hybrid ViT is an efficient approach to optimize both accuracy and speed.
- The authors put an effort into the ablation study by training the model from scratch to show that two-stage training (distillation, then fine-tuning) is critical for good performance.

Cons:
- No code for reproducibility

The manuscript is basically complete. Suggestions or deficiencies:
- "Abbreviated paper title" should be replaced with actual title
- Please publish code

---

### Decision · Program_Chairs · 2024-10-01

**Decision:**

Accept

**Comment:**

Please address the concerns of all reviewers and add testing results. Otherwise, the paper will be rejected in the last round.